# *Salmonella* Pathogenicity Island 1 (SPI-1): The Evolution and Stabilization of a Core Genomic Type Three Secretion System

**DOI:** 10.3390/microorganisms8040576

**Published:** 2020-04-16

**Authors:** Nicole A. Lerminiaux, Keith D. MacKenzie, Andrew D. S. Cameron

**Affiliations:** 1Department of Biology, Faculty of Science, University of Regina, Regina, SK S4S 0A2, Canada; lerminin@uregina.ca (N.A.L.); keith.mackenzie@uregina.ca (K.D.M.); 2Institute for Microbial Systems and Society, Faculty of Science, University of Regina, Regina, SK S4S 0A2, Canada

**Keywords:** genomic island, SPI-1, *Salmonella*, pathogenicity island, comparative genomics, type III secretion system

## Abstract

*Salmonella* Pathogenicity Island 1 (SPI-1) encodes a type three secretion system (T3SS), effector proteins, and associated transcription factors that together enable invasion of epithelial cells in animal intestines. The horizontal acquisition of SPI-1 by the common ancestor of all *Salmonella* is considered a prime example of how gene islands potentiate the emergence of new pathogens with expanded niche ranges. However, the evolutionary history of SPI-1 has attracted little attention. Here, we apply phylogenetic comparisons across the family Enterobacteriaceae to examine the history of SPI-1, improving the resolution of its boundaries and unique architecture by identifying its composite gene modules. SPI-1 is located between the core genes *fhlA* and *mutS*, a hotspot for the gain and loss of horizontally acquired genes. Despite the plasticity of this locus, SPI-1 demonstrates stable residency of many tens of millions of years in a host genome, unlike short-lived homologous T3SS and effector islands including *Escherichia* ETT2, *Yersinia* YSA, *Pantoea* PSI-2, *Sodalis* SSR2, and *Chromobacterium* CPI-1. SPI-1 employs a unique series of regulatory switches, starting with the dedicated transcription factors HilC and HilD, and flowing through the central SPI-1 regulator HilA. HilA is shared with other T3SS, but HilC and HilD may have their evolutionary origins in *Salmonella*. The *hilA*, *hilC*, and *hilD* gene promoters are the most AT-rich DNA in SPI-1, placing them under tight control by the transcriptional repressor H-NS. In all *Salmonella* lineages, these three promoters resist amelioration towards the genomic average, ensuring strong repression by H-NS. Hence, early development of a robust and well-integrated regulatory network may explain the evolutionary stability of SPI-1 compared to T3SS gene islands in other species.

## 1. Introduction

Bacterial genomes are highly dynamic, able to gain and lose genes over short evolutionary times. Comparative genomics enables the differentiation of genes that are shared by all members of a species (core genes) from the genes with variable distributions across a species (accessory genes). *Escherichia coli* is a prime example of genomic variability. Although gene content ranges from 3744 to 6844 open reading frames in individual *E. coli* isolates, only 1000 genes are shared by the over 21,000 whole genome sequences that represent this species and are currently available in GenBank [1]. As accessory genes constitute the bulk of an average bacterial genome, understanding their evolutionary histories and genetic dynamics is central to understanding bacterial functions, and capabilities.

Accessory genes often become physically linked on contiguous segments of DNA, and these islands can range from two to dozens of genes. Horizontal gene transfer (HGT) is a driver of this coalescence because a physical connection between functionally linked genes increases the frequency of successful transfer [2,3,4]. HGT disconnects the phylogenetic history of a genomic island from that of its host genome [5], which can be reflected in islands having nucleotide and codon frequencies that differ from a genomic average [6,7,8]. Further signatures of HGT are that islands often insert adjacent to genetic elements that facilitate recombination, such as tRNA genes and mobile genetic elements [8,9,10,11].

Acquisition of genomic islands can provide new ecological functions and facilitate the invasion of new niches, enabling evolutionary leaps and even speciation [12]. Examples of biological functions that mobilize as genomic islands include sugar catabolism [13], plant symbioses [14], antibiotic resistance [15], virulence factors [6], and other pathogenicity determinants [8]. The contributions of “pathogenicity islands” to bacterial evolution and niche adaptation is best understood in the model pathogen *Salmonella*. A total of 24 pathogenicity islands have been identified in this genus, though not all of these islands have been experimentally validated to contribute to virulence phenotypes [16,17,18]. The largest is *Salmonella* Pathogenicity Island 1 (SPI-1) [19], which encodes a type three secretion system (T3SS) and type three secretion effectors (T3SEs) that mediate intracellular invasion of intestinal cells in animal hosts [20,21,22,23].

Acquisition of SPI-1 is a defining event in the evolution of *Salmonella*, occurring after divergence from the common ancestor with *Escherichia* over 100 million years ago [24,25,26]. The boundaries of SPI-1 were initially determined through DNA hybridization, then later through alignment with *E. coli* DNA sequence [21,22,23] (Figure 1A). Homologous T3SS and T3SE genes have been acquired by lineages of *E. coli*, but these are rare and a prominent example in *E. coli* O157:H7 is losing functionality [27]. Broader phylogenetic comparisons have identified homologous T3SS in facultative human pathogens such as *Yersinia, Chromobacterium,* and *Shigella*, and plant pathogens such as *Pantoea* (Table 1) [28,29,30,31,32,33,34]. These T3SSs are found only in select members of each genus. For example, PSI-2 appears to be undergoing frequent HGT and loss in the genus *Pantoea* [32]. Similarly, *Yersinia* YSA is present in *Yersinia enterocolitica* but is absent from *Yersinia pestis* [35]. The Mxi-Spa T3SS has entered the *Shigella* clade multiple times on several types of pINV plasmids [36,37,38].

Despite its prominence as the archetypal pathogenicity island and intensive research attention for several decades, little is known about why SPI-1 is uniquely stable among T3SS genomic islands in Enterobacteriaceae. Previous studies of T3SS genetic architecture have included cursory analyses of the evolutionary transitions that differentiate T3SS-containing gene islands in bacteria [27,28,31,32,39,40,41,42]. Pathogenicity islands can have mosaic structures arising from the merger of smaller islands that were acquired at different points in evolutionary history [8]. While several studies have suggested that SPI-1 has a mosaic structure [43,44], an in-depth evolutionary analysis of the island’s history has not yet been conducted.

The most robust approach to find genomic islands and their boundaries is through comparative genomics. xenoGI is a recently released comparative genomics program that identifies genomic islands that are shared by a clade of bacteria as well as islands that are unique to certain strains by grouping genes by origin on a phylogenetic tree [45]. Using this locus-based approach, the objectives of our study were to: (1) identify the genomic islands of SPI-1, (2) examine the evolutionary history of the SPI-1 locus, and (3) evaluate the evolutionary history of SPI-1-encoded transcriptional regulators *hilA, hilC,* and *hilD*. Tracking how genomic islands originate, spread, and decay is key to determining how islands enable genomic diversification and adaptation. Connecting the evolutionary history of an infection-relevant pathogenicity island, SPI-1, to extensive experimental characterization of its molecular components helps develop and improve our understanding of pathogen emergence.

## 2. Materials and Methods

### 2.1. Bacterial Strains

Table 2 has a complete list of 29 bacterial strains and accession numbers included in xenoGI analysis. These strains were chosen to capture phylogenetic and GC content diversity. Unless otherwise specified, all gene and ortholog names will be as annotated as in *Salmonella enterica* serovar Typhimurium LT2 to avoid confusion over different annotations for the same genes. All genome sequence files and corresponding annotations were downloaded from NCBI Genbank. Several other strains that were used for genomic comparison but not included in xenoGI analysis are *S.* Senftenberg strain N17-509 (accession: CP026379.1), *E. coli* ISCII (accession: CBWP010000030.1), and *E. coli* O104:H11 strain RM14721 plasmid RM14721 (accession: NZ_CP027106.1). Strains with genomes at the NCBI assembly level of “complete” (gapless chromosome) were selected over draft genomes due to the higher quality and ability to distinguish independent units such as plasmids.

### 2.2. Whole-Genome Phylogenetic Tree Building for xenoGI Input

The phylogenetic tree was built using PATRIC 3.4.2 [49], which constructs trees based on coding sequence similarity. All bacteria strains listed in Table 1 were the focal group, except for *Pseudomonas aeruginosa* which was used as the outgroup (Appendix A).

### 2.3. xenoGI Parameters

Analysis was run using default parameters in xenoGI v2.2.0 [45] with the following exceptions: rootFocalClade was set to i26 (Appendix A) and evalueThresh was set to 1e-8. Computing time took approximately 3 h with 29 strains. Bed files were generated from xenoGI scripts and were used to visualize islands in Integrated Genome Browser version 9.1.0 [50] and Easyfig version 2.2.3 [51]. Analysis of islands was done with the interactiveAnalysis.py script. Annotations used to determine gene function were obtained from the Genbank flat files (.gbff) and SalCom (Table 1) [46,47,52]. Raw analysis output from xenoGI is found in Appendix A.

### 2.4. AT Content Analysis

AT content was calculated from the Genbank flat files (.gbff) for select strains in Table 1 using Geneious R11 (https://www.geneious.com) [53] and plotted as a heatmap. The average GC content for each nucleotide position was determined using a 100 base sliding window.

### 2.5. Hil Phylogenies

We used blastx 2.10.0+ [54] to search for *hil* gene homologs in other bacteria excluding the *Salmonella* clade (taxid:590) and filtering for hits that covered > 80% of the query sequence. *Salmonella enterica* serotype Typhimurium LT2 HilA (AAL21756.1), HilC (NP_461788.1), and HilD (NP_461796.1) were used as the query sequences. Because *S. enterica* Typhimurium LT2 HilC and HilD have high sequence similarity (36.4 % identity over 88 % query coverage, e-value < 3e-51, bit score = 168), we chose a 80% query cut-off filter when searching for HilC and HilD homologs to capture both regulators of interest and sufficient diversity. In other words, when searching in the *Salmonella* clade using HilC as a query with these settings, HilD would return as a hit and vice versa. The accession numbers included in the phylogenetic analysis for HilA are listed in Appendix A and accession numbers for HilC/D are listed in Appendix A. *Salmonella bongori* NCTC 12419 HilA (CCC31553.1), HilC (WP_000243993.1) and HilD (WP_000432692.1) and *Salmonella enterica* serotype Typhi CT18 HilA (CAD05983.1) were included as representatives in the phylogenetic trees. Multiple protein alignments were done using MUSCLE [55] in MEGA 7 [56]. Maximum likelihood trees were constructed using the LG+G amino acid substitution model for HilA in Figure 6, the LG+G+I amino acid substitution model for HilA in Appendix A and the JTT+G amino acid substitution model for HilC/D in MEGA X [57]. The maximum-likelihood phylogenies were supported with 1000 bootstrap replicates. Tree visualization was done with iTOL [58].

## 3. Results

### 3.1. SPI-1 is a Mosaic of Gene Islands

We conducted a fine scale analysis of gene content and architecture at the *fhlA*/-/*mutS* locus using xenoGI, a program based explicitly on phylogenetic comparisons to identify islands. Briefly, xenoGI sorts genes into families, then sorts families into islands based on synteny or location within a genome, while also accounting for amino acid similarity. Many genomic island-finding programs exist [reviewed in [62,63]], but unlike other methodologies, xenoGI requires a phylogenetic tree for input. The phylogeny is used to determine which islands are shared by a clade and to identify at which branch they were acquired [45].

The core chromosomal genes that have been previously defined as the boundaries of the SPI-1 locus are *fhlA*, encoding the formate hydrogenlyase transcriptional activator, and *mutS*, encoding a DNA mismatch repair protein, as noted previously from direct comparison of *E. coli* K-12 to *S.* Typhimurium [23,64] (Figure 1A). Comparing this locus in *Salmonella* to homologs in closely related genera *Citrobacter*, *Escherichia*, *Shigella*, *Enterobacter* and *Klebsiella* confirmed that *fhlA* and *mutS* define the boundaries of a plastic locus across these Enterobacteriaceae. In the reference *Salmonella* Typhimurium genome, xenoGI divided SPI-1 into three gene islands, which are coloured in Figure 1B: *sitABCD* (orange), *avrA-invH* (green), and *pigA*-STM2908 (purple & blue). The distinct nature of these three islands is reflected in their independent transcriptional output in infection-relevant conditions [46,48] (Figure 1C). Each of these islands is examined separately below.

### 3.2. A Cohesive SPI-1 Gene Set is Highly Conserved in *Salmonella*

To evaluate SPI-1 conservation across the genus *Salmonella*, we selected representative strains from both species (*S. enterica* and *S. bongori*), three *S. enterica* subspecies (*arizonae*, *diarizonae*, and *enterica*), and four serotypes of model pathogens (Typhimurium, Typhi, Paratyphi, and Enteritidis). This selection includes the deepest branches within the genus [65] and a range of genome sizes (4.46-5.26 Mbp). Figure 1B shows that the *avrA-invH* island (green), which encodes the T3SS and associated effectors (T3SE), is conserved across all eight *Salmonella* genomes but is absent from this locus in other Enterobacteriaceae.

A single gene in the *avrA-invH* island is not conserved across the eight representative *Salmonella* genomes. The effector gene *avrA* is absent in *S. enterica* Typhi, *S. enterica* Paratyphi and *S. enterica arizonae* (Figure 1B). The frequent loss of *avrA* in independent lineages of *Salmonella* is illustrated more comprehensively in the analysis 445 *Salmonella* strains by Worley and colleagues [66]. We note that *avrA* expression is unchanged during macrophage infection (Figure 1C), suggesting its dispensability is reflected in low integration into the SPI-1 regulatory network.

### 3.3. The *ygbA* and *sitABCD* Islands Predate Core SPI-1

The *sitABCD* island identified by xenoGI is conserved across *Salmonella*, *Citrobacter*, and *Klebsiella* (orange genes in Figure 1B). Hypothetical protein *ygbA* at this locus was classified into two islands: one in *Salmonella, Citrobacter, Escherichia* and *Enterobacter lignolyticus* and one in *Klebsiella* where is has a reverse orientation at the other side of the *sitABCD* operon (Figure 1B, blue). The simplest interpretation of this phylogeny is that the common ancestor of these Enterobacteriaceae had *ygbA* and *sitABCD* at this locus, but the *sitABCD* operon was subsequently lost from the *Escherichia/Shigella* clade and some minor reorganization has placed *ygbA* in two alternate positions either adjacent to *sitD* or *sitA*. Furthermore, the *ygbA* and *sitABCD* islands predate the insertion of the *avrA-invH* (T3SS) island in the *Salmonella* ancestor.

To test whether *E. coli* strains might encode the *sitABCD* operon at other genomic locations, we used blastx to search for orthologs in whole genome sequences. *Shigella flexneri* strain 2a 301 and *E. coli* IAI39 have *ygbA* at the *fhlA/-/mutS* locus and *sitABCD* at another location in their genome. We also examined a small number of species from sister families in the order Enterobacterales. The *sitABCD* operon is present in these bacterial families located either at chromosomal locations separate from *mutS* in *Sodalis glossinidius* and *Sodalis praecaptivus* (Pectobacteriaceae) and *Yersinia pestis* (Yersiniaceae), or on a plasmid in *Pantoea ananatis* (Erwiniaceae) (Figure 2). These species all lack *ygbA* homologs.

In *E. coli* K12, *ygbA* is the only gene located between the core genes *fhlA* and *mutS* (Figure 1A). However, the broader phylogenetic comparison in Figure 2 shows the presence of *ygbA* and *sitABCD* at this locus in the deepest-branching *E. coli* considered here (strain ISC11). This evidence further supports the supposition that many lineages of *E. coli* have lost *sitABCD* while in others the operon has relocated.

### 3.4. The Highly Variable *mutS*-Proximal Region

The region adjacent to the *mutS* gene promoter is highly variable in gene content, demonstrating gene gain and loss in all Enterobacteriaceae lineages examined here (Figure 1B). In the reference *Salmonella* genome, *S. enterica* Typhimurium, this region encodes several pathogenicity island genes (*pig* genes) and mobile elements (Figure 3). Several of the *pig* genes are regulated by SPI-1-encoded HilC, but they do not seem to contribute to *Salmonella*’s virulence phenotype and their specific functions remain unknown [43]. Reconstructing the temporal events in gene gain and loss in the representative *Salmonella* suggests an initial gain of *pigC, pigD, pphB* (purple) and a transposase (pink). This was followed by gain of *pigA, pigB* and an insertion element (blue) then loss of these elements by *S.* Enteritidis and *S.* Paratyphi. The *S.* Enteritidis and *S.* Paratyphi lineage acquired four hypothetical proteins (red) (Figure 3). No *pig* genes are found at the *fhlA*/-/*mutS* locus in other Enterobacteriaceae. *pphB* encodes a serine phosphatase that is ancestral at this locus but is located on the opposing side of *mutS* in *E. coli* (Figure 1A).

### 3.5. Decay and Loss of SPI-1

The core SPI-1 island is exceptional among related T3SS-T3SE systems in its long-term residency and stability in a bacterial clade. Nevertheless, SPI-1 can be lost, as observed in *S. enterica* serotype Senftenberg [67,68,69,70,71,72]. *S.* Senftenberg ATCC 43845 was included in the analysis presented in Figure 1B. However, the presence of pseudogenes in the *avrA-invH* island, including the SPI-1 regulator *hilD*, indicates genetic decay that is expected to cause loss of invasion functions.

To understand the genetic changes associated with the loss of SPI-1, we aligned *S.* Senftenberg ATCC 43845 to *S.* Senftenberg strain N17-509, a strain that has lost SPI-1 [72]. *S.* Senftenberg N17-509 has lost the entire SPI-1 core region (*avrA-invH*) but retains homologs to the *ygbA*, *sitABCD* and *pig* genes (Figure 4). Another gene island containing mobile elements, a toxin-antitoxin system, and a restriction endonuclease has inserted between the *pig* genes and *mutS* (Figure 4).

### 3.6. The fhlA/-/mutS Locus is a Hotspot for Island Acquisition

Our analysis shows that SPI-1 inserted at a highly plastic locus that is a hotspot for the acquisition of small and large gene islands. In *S. enterica* subsp. *diarizonae* and subsp. *arizonae*, a second island exists in each strain between *invH* and *mutS* (blue and gold, respectively, in Figure 1B). These islands consist largely of hypothetical proteins and encode mobile elements such as integrases (*S. enterica* subsp. *arizonae*) and transposases (*S. enterica* subsp. *diarizonae*).

Various *Escherichia* species and strains were included in the xenoGI analysis to capture lineage diversity (Table 2). *S. flexneri* has two integrases encoded between *ygbA* and *mutS,* but no other genes. Several *Escherichia* strains (*E. albertii, E. fergusonii* and *E. coli* IAI39) have a single transcriptional regulator *modE* located between *fhlA* and *ygbA*. *E. coli* O104:H4 2011C-3493 has a hypothetical protein between *ygbA* and *mutS*. *E. coli* K12 and *E. coli* O157:H7 are identical at this locus. Of the *Escherichia* species included in the xenoGI analysis, only *Escherichia fergusonii* had islands larger than single genes inserted at the *fhlA/-/mutS* locus; these contain genes for metabolism functions and sugar transport.

Representatives of two *Klebsiella* species, *K. oxytoca* and *K. pneumoniae*, were included in xenoGI analysis. Both strains have multiple gene islands inserted between *fhlA* and *mutS*, and two islands are shared by both species (Figure 1B, green and purple). The shared islands encode genes for an iron transport system and homologs of the sugar translocation proteins EIIB and EIIC in the phosphotransferase system, suggesting that these islands enable nutrient acquisition in *Klebsiella*.

### 3.7. AT Nucleotide Content and the Evolution of Transcriptional Control

A paradigm in bacterial genomics is that AT-rich DNA is a signature of horizontally acquired genes [73]. SPI-1 has been resident in *Salmonella* for many tens of million years, yet the island has a high AT content that has resisted amelioration to match the nucleotide composition of the core genome [65]. AT-richness of SPI-1 is maintained by a higher GC-to-AT substitution rate compared to a higher AT-to-GC substitution rate in core genes [65]. Protein-DNA interactions in gene regulatory networks may explain nucleotide frequencies that resist amelioration to genomic averages [65]. In bacteria, several global transcription factors favour the nucleotide composition and physical properties of AT-rich DNA [73,74]. One such protein is the nucleoid-associated protein H-NS, a global repressor of gene expression [75,76]. The strong repression of SPI-1 gene expression by H-NS suggest that SPI-1 resists amelioration to remain within the H-NS regulon.

We compared AT content at SPI-1 in the eight representative *Salmonella* genomes, which confirmed the expectation that all eight lineages have maintained an amazingly consistent pattern of high AT content (Appendix A). This heatmap of nucleotide composition shows the level of conservation and resilience of the high AT content of SPI-1. Moreover, it illustrates how gene islands can be internally consistent in nucleotide composition and can differ dramatically from neighbouring islands, which is especially apparent in the cases of the GC-rich islands in *Klebsiella* (Appendix A).

Two regions in SPI-1 are very AT-rich: the *hilD-hilA* and *hilC* loci (Figure 5A). This pattern fits particularly well with the model that AT content is selected to maintain membership in the H-NS regulon. Transcriptional activation of SPI-1 begins with antagonism of H-NS repression by the transcriptional activators HilC and HilD [77,78,79] (Figure 5B). When active in DNA binding, HilC and HilD activate their own promoters to create a feed-forward signal that counteracts H-NS silencing of the *hilA* promoter [80]. HilA, in turn, activates transcription of the regulator *invF* and acts directly at the T3SS and T3SE gene promoters.

### 3.8. Evolution of Transcriptional Control: Acquisition of *hilA*

HilA, a DNA binding protein in the OmpR/ToxR family of transcription factors, is the master activator of SPI-1 transcription. HilA binds to the *invF* and *prgH* promoters, triggering the activation of T3SS and T3SE genes [81,82,83]. A recent survey of T3SS in ~20,000 bacterial genomes classified SPI-1 according to gene conservation and synteny into what the authors termed category II [31]. Category II T3SS are scattered among Gammaproteobacteria and Betaproteobacteria, but *hilA* is missing from most genomes with a category II T3SS [31], suggesting this regulatory module is a relatively recent addition within this family of homologous T3SSs. We examined the evolutionary connection between *hilA*, T3SS, and T3SE genes in Proteobacteria (Figure 6, Appendix A). The HilA phylogeny does not recapitulate organismal phylogeny, consistent with the role of HGT in distributing T3SS across diverse strains and species. The dynamic architecture of SPI-1 homologs is further illustrated by aligning genes according to the largest conserved island, *spaS-invF*. This alignment helps illustrate how the T3SS and T3SE components are genetically divisible into distinct islands, *orgCBA-prgKJIH*, *hilA-iagB*, *iacP-sipADCB-sicA*, and *spaSRQPO-invJICBAEGF* (Figure 6). These constituent islands are illustrated at the bottom of Figure 6, and correspond approximately to the microsynteny blocks described by Hu and colleagues [31]: MSB1+*orgC*, MSB5+*iacP/sipA*, and MSB3+MSB4+MSB2 is island *spaSRQPO/invJICBAEGF*. The shuffled orientation and composition of these blocks means that no two genera possess the same island architecture.

*hilA* is contiguous with T3SS genes in 26 of 30 representative genomes we examined. In the other four genomes, (three strains of *Escherichia* and one strain of *Chromobacterium vaccinii*), *hilA* is either located alone on the chromosome or is absent from the genome (Figure 6). In *E. coli* O42 and *E. coli* O157:H7, an alternate transcription factor, *ygeH*, occupies the approximate location of *hilA*, adjacent to the T3SS genes (Figure 6). YgeH has low (29%) similarity to HilA, yet the *E. coli* O42 YgeH can functionally replace HilA in *Salmonella* and like HilA, its expression is regulated by H-NS [84,85].

The *E. coli* O157:H7 ETT2 is undergoing mutational attrition and becoming a cryptic gene island [27,84]. The accumulation of pseudogenes is accompanied by a loss of *hilA*, and the *ygeH* ortholog is non-functional as a transcription factor [84]. *Sodalis* presents another genus where the evolutionary stages in the decline of T3SS can be observed through comparative genomics. Members of this genus encode two T3SS homologs of SPI-1: SSR1 and SSR2. In *Sodalis praecaptivus*, SSR1 lacks *hilA* and SSR2 encodes a truncated *hilA* (Figure 6). In the endosymbiont *Sodalis glossinidius*, no *hilA* is present in the genome, and SSR1 and SSR2 are accumulating pseudogenes, consistent with a loss of island function due to the host occupying a highly specialized and obligate niche in tsetse flies.

### 3.9. Evolution of Transcriptional Control: Addition of the HilC/D Paralogs

Regulation of SPI-1 by the AraC-family proteins HilC and HilD appears to be unique because related T3SS and T3SE islands in Enterobacteriaceae do not include *hilC* or *hilD* homologs (Figure 6, Table 1). We reconstructed the evolutionary history of *hilC* and *hilD* by searching for all homologs in GenBank that are similar across 80% or more of the length of each query protein. With this search parameter, using HilC as a query recovers HilD, and vice versa, because the two proteins are closely related (36.4 % identity over 88 % query coverage, e-value < 3e-51). *hilC* and *hilD* are core elements of SPI-1 (Figure 1B), and so a GenBank search for homologs was conducted after excluding all *Salmonella* genomes from the search. Seventy-two non-redundant proteins were identified when either HilC or HilD served as a query sequence, whereas each individual query identified four unique proteins. Although HilC and HilD are ubiquitous in *Salmonella* (except for the case of *S*. Senftenberg that has lost SPI-1), searching all non-*Salmonella* genome sequences in GenBank revealed that HilC and HilD homologs are rare and sporadically distributed, occurring in only five genera outside *Salmonella* (Figure 7, Appendix A). Moreover, each genome outside of *Salmonella* has a single HilC/D homolog, highlighting another unique feature of the SPI-1 regulatory network.

A HilC/D phylogeny has low bootstrap support that prevents ordering the deepest branches (illustrated as a polytomy in Figure 7). Nevertheless, the ubiquity of HilC and HilD in *Salmonella* compared to the sporadic distribution of homologs is consistent with the two genes arising from gene duplication in *Salmonella*. The phylogeny is consistent with a series of horizontal gene transfer events that spread HilD to *Edwardsiella*, *Enterobacter*, and *Escherichia*. The largest number of homologs were detected in incomplete *Escherichia* genomes in GenBank.

In most *Escherichia*, HilC/D homologs are located adjacent to type IV pilus genes and plasmid-specific genes. For example, the complete genome of *E. coli* O104:H11 strain RM14721 includes a 106 kb plasmid, and a HilC/D homolog labeled as CofS is found on the plasmid [86] (Figure 7). This protein is part of the *cof* operon, which encodes a type 4b pilus colonization factor antigen in enterotoxigenic *E. coli* and is used to attach to host cells [87]. A homolog of HilC/D in *Enterobacter lignolyticus* was also found located near type IV pilus genes, albeit on a chromosome. A small number of HilC/D homologs were found in *Citrobacter*, *Edwardsiella*, and *Hafnia* (Figure 7). Most of these strains have complete genome sequence available, and the HilC/D homolog appears to be in the same chromosomal location across these genera. Of these genomes, only one *C. freundii* strain has a T3SS, which is located at a different position in the chromosome from the HilC/D homolog. Based on GenBank annotations, the T3SS genes are converting to pseudogenes as mutations accumulate, suggesting that this is most likely a non-functional gene island.

## 4. Discussion

In bacteria, the forces of horizontal gene transfer and recombination have significant impacts on genome content and organization, accelerating evolution and community diversity. The consequences are etched across bacterial genomes in the form of vast numbers of accessory genes and gene islands, each with distinct phylogenetic histories. Genomic islands are born from HGT: physical linkage between cooperative genes is beneficial for the simultaneous transfer of the genetic information required to assemble a cellular machine or perform a metabolic process. For example, contiguity increases the odds of successful transfer of large islands, like the genomic island replacement and serogroup conversion that drove the emergence of pathogenic *Vibrio cholerae* O139 [88].

Specific genomic regions can be hotspots for the gain and loss of islands. The plasticity of a genomic region is often attributed to the presence of DNA sequences that facilitate the integration or excision of mobile genetic elements. Prominent examples include sequences that facilitate intramolecular and intermolecular recombination such as direct repeat sequences, inverted repeat sequences, bacteriophage attachment sites, and the 3′ end of tRNA genes [9,42,89]. SPI-1 is located in a hotspot for island insertion, as revealed in our comparative analysis across multiple genera of Enterobacteriaceae (Figure 1B). However, the genetic features such as tRNA sites, phage integration sites and repeat sequence elements associated with high recombination rates are not apparent at the SPI-1 locus [23].

Even in the absence of recognizable DNA elements that facilitate recombination, *mutS* and the surrounding region are known to have high rates of horizontal exchange relative to other regions in *Salmonella* and *E. coli* genomes [43,90,91,92,93,94,95,96,97]. A defective *mutS* results in higher recombination rates, which can be beneficial in times of stress but can also have long-term negative consequences on the cell [93,94,98]. As a result of higher recombination rates, there is a more frequent changing of *mutS* alleles, which increases the likelihood that a defective *mutS* allele will be rescued by a new, functioning allele [90,93]. As the *mutS* region favours horizontal gene transfer, *Salmonella* may have concurrently acquired a *mutS* allele and a T3SS that rose to dominance in the population.

The sporadic distribution of gene islands related to SPI-1 outside the genus *Salmonella* suggests that many independent gain and loss events have scattered variant SPI-1-like islands across Proteobacteria (Figure 6). In *Shigella, Pantoea*, and *Yersinia*, T3SSs are located on plasmids, which helps explain their sporadic distribution [32,35,36,37,38,59,99]. SPI-1 has different GC content, genetic organization and phylogenetic histories than the plasmid-borne islands in *Yersinia* and *Shigella*, indicating that these plasmids were not the original source of SPI-1 in *Salmonella* [64]. Alternatively, it is unlikely that SPI-1 represents a progenitor that has been subdivided after transfer from a *Salmonella* donor to other Proteobacteria (Figure 6). *Salmonella* HilA form a distinct, highly-supported clade, providing further evidence that *Salmonella* is not the ancestral source of HilA and linked T3SS islands in other Proteobacteria.

Even archetypal islands like SPI-1 are themselves mosaics composed of smaller islands. For example, as part of the validation of xenoGI, Bush and colleagues evaluated the SPI-2 locus in *Salmonella* and found that it is composed of several smaller gene islands. SPI-2 is composed of a T3SS gene island and a tetrathionate reductase operon island [45]. These findings are consistent with a previous analysis of genetic flux in *Salmonella* that identified the *ttr* gene island as a more ancient acquisition than the SPI-2 T3SS gene island [100]. Similarly, three phylogenetically distinct T3SS islands –including homologs of SPI-1 and SPI-2– are distributed sporadically across the *Pantoea* genus, and each island is a composite of reassorting subcomponents [31,32].

During macrophage infection, the distinct transcriptional profiles of gene modules in and around SPI-1 confirms the phylogenetic evidence for the locus being an archipelago of smaller islands with distinct evolutionary histories and regulatory programs (Figure 1C). In the macrophage vacuole, transcription of *ygbA* and *sitABCD* is very high, whereas transcription of genes in the *avrA*-*invH* island is repressed. In contrast, transcription of genes STM2901-STM2908 is largely unchanged between laboratory and intracellular environments [48] (Figure 1C). Thus, the genes that contribute to shared cellular functions, such as the T3SS or metal ion import, are co-regulated. Additional examples of island-specific transcriptional responses in and around SPI-1 can be visualized in the *Salmonella* compendium of transcriptomic data [46], further confirming that the boundaries between differentially regulated transcriptional units align well with the xenoGI assignment of gene islands based on evolutionary histories.

Another aspect of the mosaicism within a genomic island is the gain and loss of transcription factors. This is pertinent in the study of SPI-1 because it encodes an unusually high number of transcription factors (five), at least four of which are required to activate transcription of SPI-1 genes. HilA is a central regulator of SPI-1 transcription, but most homologous T3SSs do not have HilA (Table 1) [31]. Some lineages in Figure 6 have lost HilA, but this is always correlated with mutational attrition of the T3SS and T3SE genes, suggesting HilA is dispensable only after its regulatory targets have lost their biological function. Although HilA is a diffusible trans-acting factor, it is almost always contiguous with the T3SS genes it regulates, further highlighting how selection for effective HGT builds and maintains the composition of genomic islands.

Xenogeneic silencing of transcription at genomic islands by H-NS and counter-silencing by transcriptional activators like HilC and HilD has been thoroughly reviewed elsewhere [73,79]. Scientists have observed that winged helix-turn-helix proteins like HilA are effective activators of horizontally-acquired AT-rich DNA because their low specificity for DNA binding sites enables competition and displacement of H-NS across broad regions of gene promoters [101]. Less well understood are the mechanisms and evolutionary steps through which local-acting dedicated transcription factors, like HilA, HilC, and HilD are gained and integrated into a functionally cohesive island. We posit that the unusually long-term stability of SPI-1 is *Salmonella* arises from the fine-tuning of transcriptional activation only when ecologically appropriate. Further, we suspect that integration is reinforced by coordination of SPI-1 expression with core housekeeping functions by regulators like H-NS, in addition to development of cross-talk between SPI-1 regulators and the transcriptional control of other genomic islands [102]

SPI-1 has many features consistent with the classical model of a genomic island: a history of insertion revealed by comparative genomics plus a high AT content exceeding the genomic average. Yet in other respects, SPI-1 is unusual as a genomic island. For example, unlike its related islands that demonstrate short residency times in bacterial strains, SPI-1 has been resident in its host for many tens of millions of years. The average residency time of a genomic island is difficult to estimate due to the absence of a calibrated evolutionary record in the vast majority of bacteria. In laboratory conditions, pathogenicity islands have natural deletion frequencies ranging from 10^−4^ to 10^−7^ [reviewed in [103]], including several pathogenicity islands (PAIs) in uropathogenic *E. coli* [104] and the High-Pathogenicity Island (HPI) in *Yersinia* [105]. Residency and lasting integration will depend on organismal biology, stochastic molecular genetic events, and broader ecological pressures acting on the host organism and the host’s ability to use the genetic potential in the island.

The *sitABCD* genes are traditionally treated as members of the SPI-1 island. Yet they have been previously been suggested to have an alternate history than the other SPI-1 genes based on the similarity of their AT content to the genomic average, which contrasts with the AT-rich SPI-1 [44] (Appendix A). Our analysis supports this hypothesis, as the *sitABCD* genes are found at the same locus in *Klebsiella* and at other locations in other Enterobacteriaceae (Figure 2). There are conflicting reports on whether *sitABCD* is essential for SPI-1 virulence, but the majority argue that is it required for infection [44,106,107,108]. Although *sitABCD* may be useful for iron and/or manganese acquisition during infection, this operon was present at the *fhlA/-/mutS* locus prior to acquisition of the structural T3SS genes. If *sitABCD* and SPI-1 do cooperate during infection, their physical proximity on the genome may be purely coincidence.

xenoGI is an easy-to-use program that was able to conduct our multi-genome analysis in under three hours. Basing comparative analysis on a phylogenetic tree makes xenoGI a powerful tool to analyze the history of genomic islands. When released, xenoGI was validated with several examples using similar clades of bacteria to our analysis (*Salmonella* and *Escherichia*, with *Klebsiella* and *Serratia* used as outgroups) [45]. Our analysis covers a larger phylogenetic breadth than previously tested, and xenoGI resolved *sitABCD*, *fhlA*, and *mutS* gene conservation consistent with the whole genome phylogenies. xenoGI is constrained to the analysis of coding sequences from complete genomes, meaning that it is unable to recognize small RNAs and promoter sequences. There is a small RNA, *invR*, adjacent to *invH* that is an important regulator of SPI-1 [109], but it was not considered in this analysis for this reason.

SPI-1 can almost be considered a core gene set in *Salmonella*, but it fails the bioinformatic definition of a “core” genetic element due to its loss from some members of the genus. Similar to the loss from strains of *S.* Senftenberg [67,68,69,70,71,72], isolates of *S.* Litchfield have also lost SPI-1 [68,71]. These strains were isolated from environmental samples, not from animals or human infections [68,71]. These strains remain able to invade animal cells, albeit at a reduced rate [69]. After many millions of years of integration and a near ubiquity in extant members of the diverse *Salmonella* genus, SPI-1 is expected to perform key ecological functions, including in the less-studied species *S. bongori*. The natural loss of SPI-1 presents a test case for predicting ecological functions based on gene content [110]. Specifically, a reduced ability to colonize animal hosts may be accompanied by a loss of metabolic pathways for host-derived nutrients. Loss of SPI-1 may not be strongly selected against in some niches, but the lineages lacking SPI-1 may be evolutionary dead ends [111].

## Figures and Tables

**Figure 1 microorganisms-08-00576-f001:**
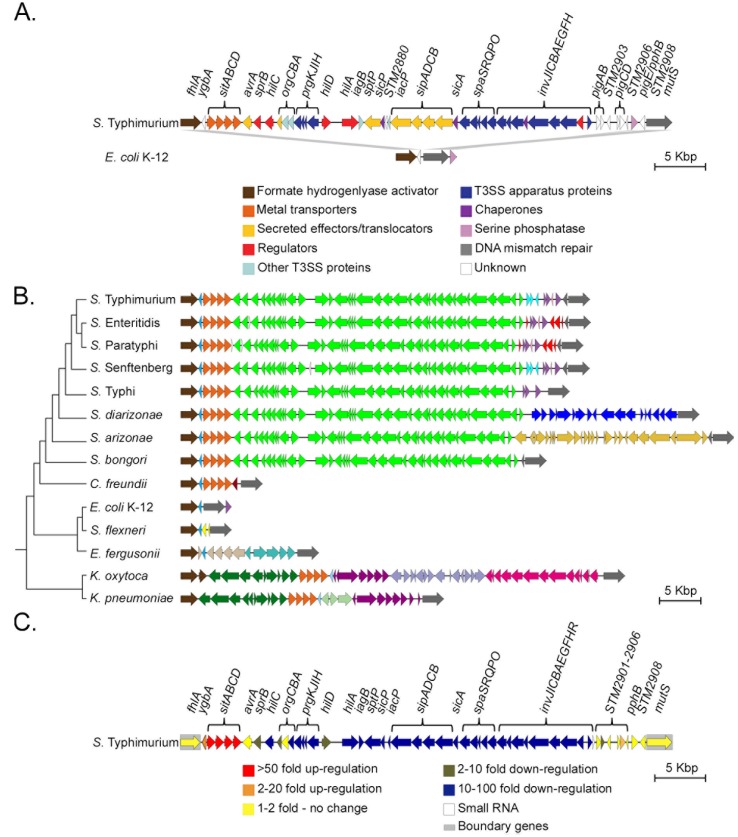
Comparative genomic analysis of *Salmonella* Pathogenicity Island 1 (SPI-1). (**A**) Alignment of *Salmonella enterica* serovar Typhimurium LT2 SPI-1 to the same locus in *Escherichia coli* K12. Genes are coloured by function based on annotations in Genbank and SalCom [46,47]. Grey bars represent sequence homology determined by blastx (minimum length 100 bp, e-value < 0.00001). (**B**) Alignment of the SPI-1 locus in *Salmonella, Escherichia, Citrobacter,* and *Klebsiella.* Whole-genome phylogeny was constructed with PATRIC. Gene colouring corresponds to different gene clusters identified by xenoGI [45]: *fhlA* (dark brown), *ygbA* (blue), *sitABCD* (orange), *avrA – invH* (green), *pig* genes and *pphB* (light blue/purple) and *mutS* (grey). *S. enterica* Senftenberg strain ATCC 43845 is included in this figure. Small open reading frames (white) between *sitD* and *hilA* in *S. enterica* Enteriditis, *S. enterica* Paratyphi, *S. enterica* Senftenberg may be due to different annotation pipelines and will not be examined here. (**C**) The transcriptional response to the intra-macrophage environment by *S.* Typhimurium 4/74 reflects that SPI-1 is an island composed of transcriptionally-cohesive modules. Genes are coloured according to the fold differences quantified by RNA-seq in the macrophage vacuole compared to early stationary phase in Lennox broth; data from [48].

**Figure 2 microorganisms-08-00576-f002:**
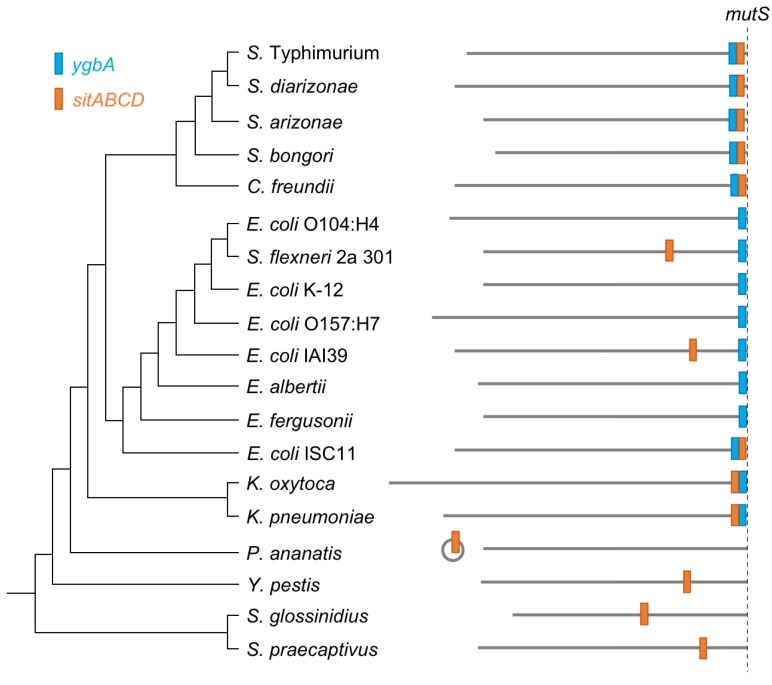
Genomic positions of *ygbA* and *sitABCD* relative to *mutS* in select Enterobacterales. The whole-genome phylogeny was constructed by PATRIC [49]. Variations of the *ygbA* and *sitABCD* gene clusters are coloured for *ygbA* (blue) and *sitABCD* (orange). Chromosome lengths are drawn to scale.

**Figure 3 microorganisms-08-00576-f003:**
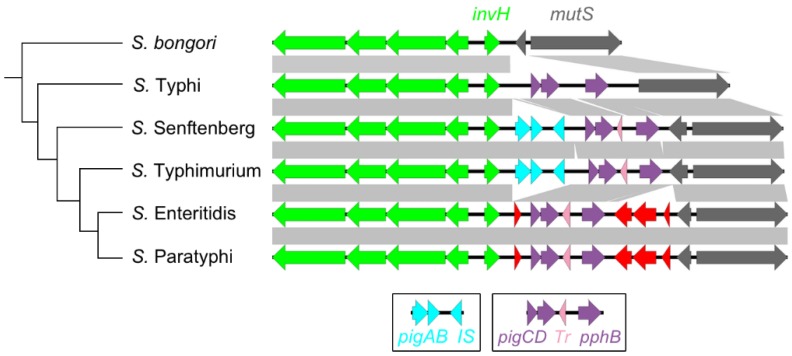
Alignment of the SPI-1 downstream boundary across the *Salmonella* clade. Whole-genome phylogeny was constructed with PATRIC [49]. Gene colouring corresponds to different gene clusters identified by xenoGI [45]: *invAEGFH* (green), *pigAB* and insertion element (blue), *pigCD* (purple), transposase (pink), *pphB* (purple), hypothetical coding sequences (red, grey). *S. enterica* Senftenberg strain ATCC 43845 is included in this figure. Grey bars represent sequence homology determined by blastx (min length 100 bp, e-value < 0.00001). IS, insertion element; Tr, transposase.

**Figure 4 microorganisms-08-00576-f004:**
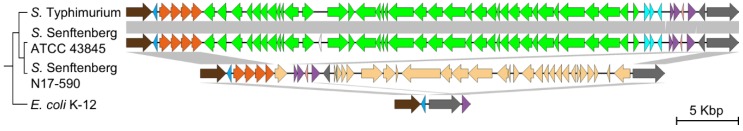
Alignment of the SPI-1 locus in *S. enterica* Typhimurium LT2, *S. enterica* Senftenberg strains ATCC 43845 and N17-590 and *Escherichia coli* K12. Gene colouring corresponds to different gene clusters identified by xenoGI [45]: *fhlA* (dark brown), *ygbA* (blue), *sitABCD* (orange), *avrA – invH* (green), *pig* genes and *pphB* (light blue/purple) and *mutS* (grey). Genomic island in *S.* Senftenberg N17-509 with no homology to SPI-1 is coloured beige. Grey bars represent sequence homology determined by blastx (min length 100 bp, e-value < 0.00001).

**Figure 5 microorganisms-08-00576-f005:**
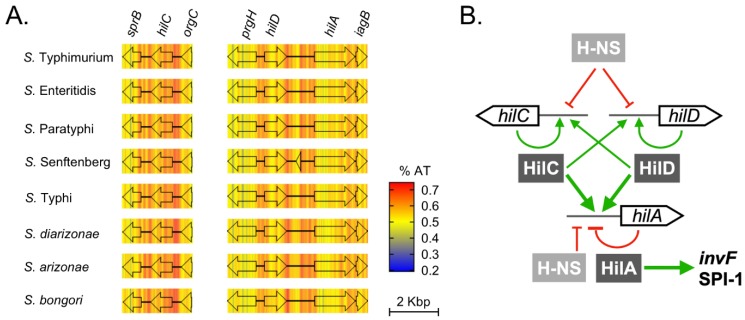
Nucleotide composition and gene regulation in SPI-1. (**A**) AT content schematic of *hil* gene promoters across eight *Salmonella* strains. AT content was overlaid as a heatmap on the regions surrounding *hilC*, *hilD* and *hilA*. Heatmap values were generated with a 100 base sliding window using Geneious R11 [53]. (**B**) Core elements of the SPI-1 transcriptional regulatory network.

**Figure 6 microorganisms-08-00576-f006:**
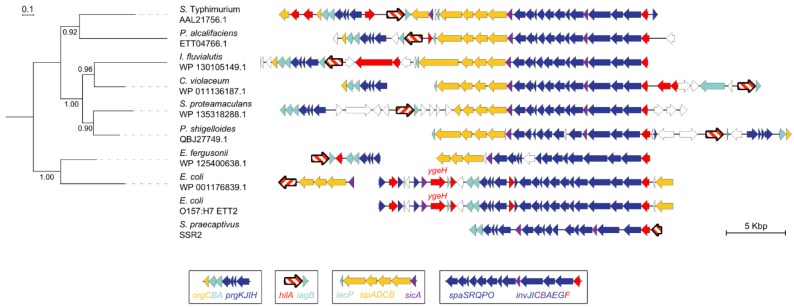
HilA phylogeny and genetic context for *hilA* in T3SS islands. Best blastx hits covering at least 80% of the query (*S. enterica* Typhimurium LT2 HilA) were identified and one representative protein sequence was selected from each species to capture phylogenetic diversity. Protein sequences were aligned using MUSCLE [55] and phylogeny was built using a maximum-likelihood model LG+G with 1000 bootstrap replicates [57]. *hilA* is illustrated in bold with white and red hashes and genes are coloured according to function (see Figure 1 legend). Conserved gene modules in T3SSs with *hilA* are identified at the bottom of the figure. Two strains (bottom) were included in xenoGI analysis but not the phylogenetic tree. Both have a T3SS but are either missing *hilA* (*E. coli* O157:H7 Sakai) or have a truncated *hilA* (*S. praecaptivus*). For a more detailed reconstruction of *hilA* evolution, see Appendix A.

**Figure 7 microorganisms-08-00576-f007:**
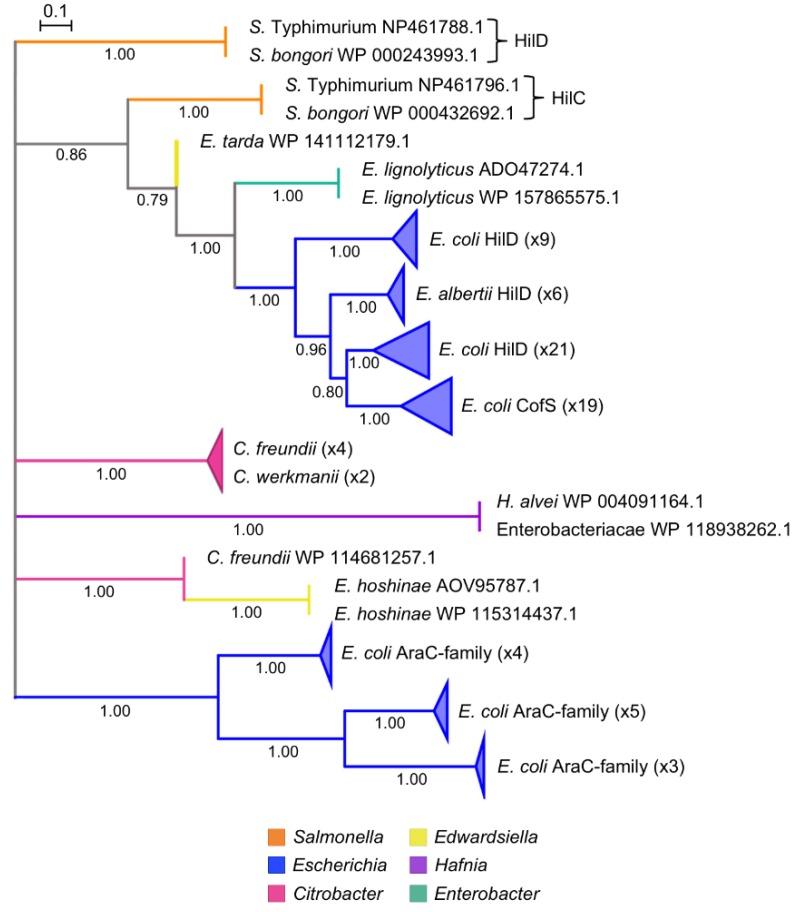
HilCD phylogeny. Nodes with bootstrap support below 0.70 are condensed to polytomies, and branches are coloured by genus. The number of strains in each collapsed node is indicated on the figure. *S. bongori* and *S. enterica* Typhimurium HilC and HilD sequences were included as representatives from the *Salmonella* clade. The top 100 best blastx hits covering at least 80% of the query for *S. enterica* Typhimurium LT2 HilC and HilD sequences were aligned using MUSCLE [55] and the phylogeny was inferred using a maximum-likelihood model JTT+G with 1000 bootstrap replicates [57]. For the full unrooted phylogeny, see Appendix A.

**Table 1 microorganisms-08-00576-t001:** Nomenclature of genes in T3SSs related to SPI-1 ^1^.

*Salmonella* SPI-1	*Pantoea* PSI-2	*Shigella* Mxi-Spa	*Escherichia* ETT2	*Escherichia* LEE	*Yersinia* YSA	*Yersinia* YSC	*Sodalis* SSR1	*Sodalis* SSR2	*Pseudomonas* Psc/Pcr/Pop/Exs	*Chromobacterium* CPI-1	Universal ^2^	Function ^3^
*avrA*						*yopP*						Effector
*sprB*												Regulator
*hilC*		*S0103*										Regulator
*orgC*										*corC*		Effector
*org*B		*mxiL/mxiN*	*orgB*	*escL*	*YE3555*	*yscL*		*orgAb*	*pscL*	*corB*	*sctL*	Needle assembly
*orgA*		*mxiK*	*orgA*		*YE355*4	*yscK*		*orgA*	*pscK*	*corA*	*sctK*	Needle assembly
*prgK*	*psaJ*	*mxiJ/yscJ*	*eprK*	*escJ*	*ysaJ*	*yscJ*	*ysaJ*	*prgK*	*pscJ*	*cprK*	*sctJ*	Inner mb ring
*prgJ*	*psaI*	*mxi*I	*eprJ*	*escI*	*ysaI*	*ysc*I	*ys*aI	*prgJ*	*pscI*	*cprJ*	*sctI*	Needle subunit
*prgI*	*psaG*	*mxiH*	*eprI*	*esc*F	*YE3551*	*yscF*	*ysaG*	*prgI*	*pscF*	*cprI*	*sctF*	Needle subunit
*prg*H	*psaF*	*mxiG*	*eprH*	*escD*	*YE3550*	*yscD*	*ys*aF	*prgH*	*pscD*	*cprH*	*sctD*	Inner mb ring
*hilD*												Regulator
*hilA*			*ygeH*					*hilA*		*cilA*		Regulator
*iagB*	*psaH*	*ipgF*	*ipgF*	*etgA*	*ysaH*		*ysaH*			*iagB*		Transglosylase
*sptP*					*yspP*	*yop*H				*CV0974*		Effector
*sicP*	*psa7*											Chaperone
*iacP*	*psaC/acpM*	*ipg*G			*acpY*		*ysaC*			*iac*P		Acyl carrier
*STM2880*												Unknown
*sipA*		*ipaA*			*yspA*		*ysp*A			*cipA*		Effector
*sipD*	*psp*D	*ipaD*		*espA*	*yspD*	*lcrV*	*yspD*		*pcrV*	*cipD*		Tip complex
*sipC*	*pspC*	*ipaC*		*espB*	*yspC*	*yopD*	*yspC*		*popD*	*cip*C		Effector
*sipB*	*pspB*	*ipaB*		*espD*	*yspB*	*yopB*	*yspB*		*popB*	*cip*B		Effector
*sicA*	*pchA*	*ipgC*	*ygeG*	*cesD*	*sycB*	*syc*D	*sycB*	*sicA*	*pcrH*	*cicA*		Chaperone
*spaS*	*psaU*	*spa40*	*epaS*	*escU*	*ysaU*	*yscU*	*ysaU*	*spaS*	*pscU*	*cpaS*	*sctU*	Export apparatus
*spaR*	*psaT*	*spa29*	*epaR*	*escT*	*ysaT*	*yscT*	*ysaT*	*spaR*	*pscT*	*cpaR*	*sctT*	Export apparatus
*spaQ*	*psaS*	*spa9*	*epaQ*	*escS*	*ysaS*	*yscS*	*ysaS*	*spaQ*	*pscS*	*cpaQ*	*sctS*	Export apparatus
*spaP*	*psaR*	*spa24*	*epaP*	*escR*	*ysaR*	*yscR*	*ysaR*	*spaP*	*pscR*	*cpaP*	*sct*R	Export apparatus
*spaO*	*psaQ*	*spa33*	*epaO*	*escQ/sepQ*	*ysaQ*	*yscQ*	*ysaQ*	*spaO*	*pscQ/hrcQ*	*cpaO*	*sctQ*	Cytoplasmic ring
*invJ*	*psaP*	*spa32/spaN*	*eivJ*	*escP/orf1*6	*yspN*	*yscP*	*ysaP*	*spaN*	*pscP*	*cpaN*	*sctP*	Needle assembly
*invI*	*psaO*	*spa13/spaM*	*eivI*	*escO/escA/orf15*	*YE3543A*	*yscO*	*ysaO*	*spaM*	*psc*O	*cpaM*	*sctO*	Needle assembly
*invC/spaL*	*psaN*	*spa47/spaL*	*eivC*	*escN*	*ysaN*	*yscN*	*ysa*N	*invC*	*pscN*	*civC*	*sctN*	ATPase
*invB*	*psaK*	*spa15/spaK*			*ysaK*		*ysa*K	*invB*		*civB*		Chaperone
*invA*	*psaV*	*mxiA*	*eivA*	*esc*V	*ysaV*	*yscV/lcrD*	*ysaV*	*invA*	*pcrD*	*civA*	*sctV*	Export apparatus
*invE*	*psaW*	*mxiC*	*eivE*	*sepL/sepD*	*ysaW*	*yopN/tyeA*	*ysaW*	*invE*	*popN*	*civE*	*sctW*	Export regulator
*invG*	*psaC*	*mxiD*	*eivG*	*escC*	*ysaC*	*yscC*	*ysaC*	*invG*	*pscC*	*civG*	*sctC*	Outer mb ring
*inv*F	*mxiE/ysaE*	*mxiE*	*eiv*F		*ysaE*	*virF*	*ysaE*	*invF*	*exsA*	*civF*		Regulator
*invH*		*mxiM*				*yscW*			*exsB*			Export apparatus
*pigA*												Unknown
*pigB*												Unknown
*STM2903*												Insertion element
*pigC*												Unknown
*pigD*												Unknown
*STM2906*												Transposase
*pphB*												Phosphatase
*STM2908*												Unknown

^1^ Gene names curated for Pantoea PSI-2 [32,40], Shigella Mxi-Spa [39,41,59], Escherichia ETT2 Sakai [27], Escherichia EPEC LEE [39], Yersinia YSA [30,40,41,60,61], Yersinia YSC [39,42], Sodalis SSR1 [29,40], Sodalis SSR2 [29], Pseudomonas aeruginosa Psc/Pcr/Pop/Exs [39,42], Chromobacterium CPI-1 [28]. ^2^ Universal gene names from [31,52]. ^3^ Functions from [46,47,52]. Mb = membrane.

**Table 2 microorganisms-08-00576-t002:** List of bacterial strains and genome assembly versions used in xenoGI analysis.

Genus	Species	Strain	% GC	Mbp	Genbank Accession	Genbank Assembly Version
*Citrobacter*	*freundii*	CFNIH1	52.2	5.09	NZ_CP007557.1	GCA_000648515.1_ASM64851v1
*Citrobacter*	*koseri*	ATCC BAA-895	53.8	4.72	NC_009792.1	GCA_000018045.1_ASM1804v1
*Enterobacter*	*cloacae* subsp. *cloacae*	ATCC 13047	52.47	5.32	NC_014121.1	GCA_000025565.1_ASM2556v1
*Enterobacter*	*lignolyticus*	SCF1	57.2	4.81	NC_014618.1	GCA_000164865.1_ASM16486v1
*Escherichia*	*albertii*	KF1	49.7	4.7	NZ_CP007025.1	GCA_000512125.1_ASM51212v1
*Escherichia*	*coli*	O157:H7 Sakai	50.45	5.59	NC_002695.1	GCA_000008865.1_ASM886v1
*Escherichia*	*coli*	K12 MG1655	50.8	4.64	NC_000913.3	GCA_000005845.2_ASM584v2
*Escherichia*	*coli*	IAI39	50.6	5.13	NC_011750.1	GCA_000026345.1_ASM2634v1
*Escherichia*	*coli*	O104:H4 str 2011C-3493	50.63	5.44	NC_018658.1	GCA_000299455.1_ASM29945v1
*Escherichia*	*fergusonii*	ATCC 35469	49.88	4.64	NC_011740.1	GCA_000026225.1_ASM2622v1
*Klebsiella*	*pneumoniae* subsp. *pneumoniae*	HS11286	57.14	5.68	NC_016845.1	GCA_000240185.2_ASM24018v2
*Klebsiella*	*oxytoca*	CAV1374	55.26	7.23	NZ_CP011636.1	GCA_001022195.1_ASM102219v1
*Pantoea*	*agglomerans*	IG1	55.02	5.12	NZ_CP016889.1	GCA_001709315.1_ASM170931v1
*Pantoea*	*ananatis*	LMG 5342	53.4	4.14	NC_016816.1	GCA_000283875.1_ASM28387v1
*Pantoea*	*stewartii*	DC283	54.2	4.53	NZ_CP017581.1	GCA_002082215.1_ASM208221v1
*Pseudomonas*	*aeruginosa*	PAO1	66.6	6.26	NC_002516.2	GCA_000006765.1_ASM676v1
*Salmonella*	*bongori*	NCTC 12419	51.3	4.46	NC_015761.1	GCA_000252995.1_ASM25299v1
*Salmonella*	*enterica* subsp. *enterica* Enteritidis	P125109	52.2	4.69	NC_011294.1	GCA_000009505.1_ASM950v1
*Salmonella*	*enterica* subsp. *enterica* Typhi	CT18	51.88	5.13	NC_003198.1	GCA_000195995.1_ASM19599v1
*Salmonella*	*enterica* subsp. *enterica* Typhimurium	LT2	52.22	4.95	NC_003197.2	GCA_000006945.2_ASM694v2
*Salmonella*	*enterica* subsp. *enterica* Paratyphi *C*	RKS4594	52.21	4.89	NC_012125.1	GCA_000018385.1_ASM1838v1
*Salmonella*	*enterica* subsp. *enterica* Senftenberg	ATCC 43845	51.93	5.26	CP019194.1	GCA_000486525.2_ASM48652v2
*Salmonella*	*enterica* subsp. *arizonae*	62z23 RSK2980	51.4	4.6	NZ_CP006693.1	GCA_000018625.1_ASM1862v1
*Salmonella*	*enterica* subsp. *diarizonae*	HZS154	51.4	5.09	CP023345.1	GCA_002794415.1_ASM279441v1
*Shigella*	*flexneri*	2a 301	50.67	4.83	NC_004337.2	GCA_000006925.2_ASM692v2
*Sodalis*	*glossinidius*	mortisans	54.51	4.29	NC_007712.1	GCA_000010085.1_ASM1008v1
*Sodalis*	*praecaptivus*	HS1	57.13	4.29	NZ_CP006569.1	GCA_000517425.1_ASM51742v1
*Yersinia*	*enterocolitica* subsp. *enterocolitica*	8081	47.25	4.68	NC_008800.1	GCA_000009345.1_ASM934v1
*Yersinia*	*pestis*	CO92	47.61	4.83	NC_003143	GCA_000009065.1_ASM906v1

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
