# Peer review of "Salmonella Pathogenicity Island 1 (SPI-1): The Evolution and Stabilization of a Core Genomic Type Three Secretion System"

_microorganisms, 2020, doi:10.3390/microorganisms8040576_

Round 1

Reviewer 1 Report

This bioinformatics study adds to similar previous studies by expanding on more bacteria using mainly the xenoGI software package for the pathogenicity island SPI-1, best known in Salmonella. It is a descriptive gene-centric approach with some comparisons and discussion on the sit genes and hilACD regulator genes.

The following points need to be addressed:

  • Line 74: Table 2 is presented in the text before Table 1
  • Lines 82-89 are repeated from above: delete.
  • 1 is shown long before it is presented in the text. Also, in panel C, the color choices in the reds do not distinguish clearly 10-50 and 50-100x upregulations.
  • Table 1 lists the functions of the gene products without discussing them. Moreover, since similar tables of orthologous genes have already been published for most of the listed bacteria, the table could be moved to the supplemental material.
  • Lines 153-161: The authors focused on genes between fhlA and mutS, obviously a target for HGT, whether for the SPI-1 gene cluster or gene clusters from other Enterobacteriaceae. It would have been relevant to analyze the DNA sequence of this intergenic region which makes it a target for HGT or discuss what was published about this and put it in the context of the current study. Lines 397-400 makes a statement that was not studied in detail; e.g. do none of the thousands of Enterobacteriaceae genomes suggest anything special for the fhlA-mutS intergenic DNA sequences (inverted repeats, known tRNA site, phage DNA att sites, etc…)?
  • Lines 170-181: This sounds more like a review of published data and should be shortened since nothing new is presented. The information in these paragraphs would be better suited for the Introduction or Discussion, not the Result section.
  • Lines 196-201: Since E. coli is older than Salmonella from an evolutionary point of view, and since the sit genes are elsewhere than between fhlA and mutS, couldn’t genomic rearrangement have originated the pathogenicity island, with the T3SS genes coming in after the sit genes? Also, the presence of T3SS on plasmids in other Enterobacteriaceae could explain Salmonella T3SS gene clusters acquisition by conjugation, plasmid integration and chromosomal rearrangements.
  • 6 legend: The gene modules are at the bottom, not top of the figure.
  • Line 360: What is a “complete genome”?
  • Line 367: “HilC/D has therefore evolved to regulate type IV pili” suggests that the HilC/D genes of Salmonella are older than the “orthologues” or similar genes regulating pili. Isn’t the opposite more likely or at least as likely?
  • Lines 410-411: This is not clear. The first part of this sentence needs to be reformulated.

Reviewer 2 Report

In their study, the authors carried out analysis of the genomic evolution and stabilization of SPI-1 fragment, T3SS, trying to explain evolutionary stability of SPI-1 versus T3SS islands in other species.
Salmonella Pathogenicity Island 1 (SPI-1) is the largest and most intensive studied fragment of Salmonella genome, and its major function was proven to mediate intestinal cell invasion during Salmonella infection.
Authors provide lot of interesting analysis of SPI-1 plasticity and regulation throughout the Enterobacteriaceae family. I find this publication as an interesting and elegant work, with plenty of useful analysis genetic elements like pathogenicity islands, especially in wider context of phylogeny.

Minor comments:

1)
# 72 Materials and Methods Lines form #72 to #81 were repeated in Lines #82-89

2) I would like to see 2-3 sentences in Introduction part in which Authors explain their main goal.
